# Passive Biotelemetric Detection of Tibial Debonding in Wireless Battery-Free Smart Knee Implants

**DOI:** 10.3390/s24051696

**Published:** 2024-03-06

**Authors:** Thomas A. G. Hall, Frederic Cegla, Richard J. van Arkel

**Affiliations:** 1Biomechanics Group, Department of Mechanical Engineering, Imperial College London, London SW7 2AZ, UK; t.hall@imperial.ac.uk; 2Non-Destructive Evaluation Group, Department of Mechanical Engineering, Imperial College London, London SW7 2AZ, UK; f.cegla@imperial.ac.uk

**Keywords:** biomedical telemetry, orthopaedics, arthroplasty, bone cement, aseptic loosening, remote patient monitoring, ultrasound, wireless sensor

## Abstract

Aseptic loosening is the dominant failure mechanism in contemporary knee replacement surgery, but diagnostic techniques are poorly sensitive to the early stages of loosening and poorly specific in delineating aseptic cases from infections. Smart implants have been proposed as a solution, but incorporating components for sensing, powering, processing, and communication increases device cost, size, and risk; hence, minimising onboard instrumentation is desirable. In this study, two wireless, battery-free smart implants were developed that used passive biotelemetry to measure fixation at the implant–cement interface of the tibial components. The sensing system comprised of a piezoelectric transducer and coil, with the transducer affixed to the superior surface of the tibial trays of both partial (PKR) and total knee replacement (TKR) systems. Fixation was measured via pulse-echo responses elicited via a three-coil inductive link. The instrumented systems could detect loss of fixation when the implants were partially debonded (+7.1% PKA, +32.6% TKA, both *p* < 0.001) and fully debonded in situ (+6.3% PKA, +32.5% TKA, both *p* < 0.001). Measurements were robust to variations in positioning of the external reader, soft tissue, and the femoral component. With low cost and small form factor, the smart implant concept could be adopted for clinical use, particularly for generating an understanding of uncertain aseptic loosening mechanisms.

## 1. Introduction

Arthroplasty is a restorative, pain-relieving treatment for end-stage knee osteoarthritis [1]. Whilst the treatment is widely practised and highly successful, its high volume use (>2 million p.a.) means that large numbers of patients are affected by implant failures even when percentage revision rates are low [2,3]. Aseptic loosening has been the principal failure mechanism since 2016, accounting for one in three of all revision procedures [4]. Timely detection of loosening will not only avoid its most severe, life-changing consequences, but by limiting progressive damage, treatment outcomes will also be improved.

Aseptic loosening may be induced by one or more of several biological and mechanical mechanisms [5]. Historically, the predominant cause of loosening was biological [6]: polyethylene wear debris caused osteolysis and resorption of periprosthetic bone. Following improvements in implant materials and manufacturing, however, osteolysis rates have substantially reduced. Instead, most contemporary loosening failures are mechanical in origin, initiating from the cement mantle. A bias towards failure at the implant–cement interface (tibial debonding) has been regularly reported in recent arthroplasty literature [7,8,9,10,11,12,13,14,15,16,17,18]. Indeed, a 94% failure rate at the implant–cement interface was recorded in a recent retrospective study of 149 knees that were revised for aseptic loosening [8]. Whilst several causal factors have been implicated in this failure mode, including cementation technique [19,20,21], implant design [10], and surface finish [22], no common systematic mechanism or presentation has been reported [23].

Suspected aseptic loosening is conventionally examined using radiographs, which are effective in detecting gross loosening through implant malposition, radiolucency, or periprosthetic fracture. Radiographic examination, however, is either poorly sensitive to the early stages of aseptic loosening [9,24] or poorly specific in determining implant stability [25] and cannot be used alone to differentiate between septic and aseptic cases [26]. Indeed, aseptic loosening is often only confirmed intra- or postoperatively, with the preoperative assessment frequently being a misdiagnosis of a low-grade infection [27]. Several radionuclide imaging techniques—scintigraphic [28,29] and tomographic [30,31,32]—have been implemented to provide more conclusive diagnoses of aseptic loosening [24]; radionuclide imaging has also been used to detect periprosthetic joint infection with reasonable success, but these advanced imaging techniques are time-consuming and only available at specialist centres.

As an alternative to radiological investigations, which can only be undertaken by specialist clinicians following patient self-reporting, a “smart implants” approach functionalises prostheses with failure-detecting mechanisms for simpler, more prompt diagnoses. The field of orthopaedics has been using sensor-embedded implants for several decades to generate valuable information for orthopaedic research, and concepts have been presented to detect loosening through analysis of vibration [33], micromotion [34,35], and acoustic response [36], but such implants have not yet translated into mainstream clinical practice. One of the challenges faced by designers of smart implants has been minimising the overall size and cost of the onboard circuitry used for power, communication, sensing, and processing, with inflated architectures having been associated with increased bone loss [37]. In particular, batteries have limited power reserves and increase device size, but they can be offloaded when using passive interrogation techniques.

In our previous work [36], a concept for loosening detection requiring only two implanted components (a piezoelectric transducer and a coil) was demonstrated in benchtop tests. The aim of this work was to embody the system into modern knee replacement systems and comprehensively characterise measurement sensitivity for detecting tibial debonding. A partial knee prosthesis was analysed as it is one of the smallest cemented implants in routine clinical use, and to date, no smart implant system has been small enough to be successfully embodied within such a device. A total knee replacement system was also analysed, as it is the highest volume cemented orthopaedic procedure.

## 2. Materials and Methods

Partial and total knee replacement systems were made “smart” with minimal embedded circuitry, with fixation measurements to detect tibial debonding elicited using an external reader.

### 2.1. Smart Implant Design

The smart partial knee replacement (Figure 1a) was a modified version of an Oxford Fixed Lateral Partial Knee (Zimmer Biomet, Warsaw, IN, USA). It consisted of a standard femoral component (CoCr) and an instrumented fixed-bearing tibial component (UHMWPE/CoCr). The ultrasonic sensing element was a shear-wave piezoelectric transducer (material: NCE51; area: 5 × 5 mm^2^; resonant frequency: 3.5 MHz; Noliac, Kvistgaard, Denmark) affixed to the superior surface of the metallic subcomponent in a posterolateral position (Figure 1b). The embedded coil was wound around the perimeter of the overmoulded polymer subcomponent (external groove: 1 × 1 mm^2^) for inductive coupling to an external reader.

The smart total knee replacement was a modified version of a Persona Total Knee (Zimmer Biomet, Warsaw, IN, USA). Similarly, it constituted an unmodified femoral component (CoCr) and an instrumented fixed-bearing tibial component (UHMWPE/Ti); a piezoelectric transducer of the same specification was affixed in a medial position using the same epoxy technique, and the embedded coil (10 turns) was wound around the perimeter of the polymer bearing in a groove of the same dimensions.

The same external reader (Figure 1c) was used to acquire measurements from both implants. The reader consisted of a three-turn circular-coil transmitter (diameter: 105 mm) and a twelve-turn helical-coil receiver (diameter: 105 mm; height: 10 mm), which were used to initiate and receive a pulse-echo measurement (Figure 1d) with five cycles at a center frequency of 3.5 MHz.

### 2.2. Signal Acquisition and Processing

Raw signals were initiated by a five-cycle, Hanning-windowed toneburst (central frequency: 3.5 MHz), and the response was sampled at 100 MHz (Handyscope HS5, TiePie, Sneek, The Netherlands). These signals were amplified (gain: 60 dB; WaveMaker Duet, Macro Design Ltd., London, UK).

Data were then analysed in MatLab (R2018b, MathWorks) with a custom script. The data were averaged across many measurements (N > 5000), bandpass-filtered (passband: 2–5 MHz), resampled at 10 GHz, and cross-correlated with the original signals. Hilbert envelopes were extracted from the processed signals, and the fixation measurement (Γ) was computed as the arithmetic average of the relative amplitude between successive echoes for the first three echo pairs.

### 2.3. Simulated Tibial Debonding

Prior to cementation, n = 50 measurements were acquired. The instrumented tibial and unmodified femoral components of both implants were then cemented into synthetic knee models using polymethyl methacrylate (PMMA) bone cement (target layer thickness: ~2 mm). The knee models (Figure 1a) consisted of synthetic tibia and femur (Sawbones 1146, Vashon Island, WA, USA) plus string lateral collateral, medial collateral, and posterior cruciate ligaments. The models were cut using standard instrumentation and surgical techniques. A new bone model was used for each implantation. Bone cement was mixed in a 1.5:1 mass-volume ratio of powder to liquid and applied whilst viscous to ensure good interdigitation at the cement-bone interface. Measurements (n = 50) were acquired post-cementation whilst the implants were “well-fixed”.

The implants were then loosened at the implant–cement interface with a manually applied anterior lift-off moment. Measurements (n = 50 per state) were acquired in each of three different states of loosening: partially debonded (loose directly beneath the transducer but still fixed more posteriorly), fully debonded with no implant migration (in situ), and fully debonded with displacement from the original position.

In each measurement condition, the embedded and reader coils were coplanar with the reader coil axis parallel to the tibial proximal–distal z axis (Figure 2). The femoral component was moved through a range of flexion angles (0–90°) for varying ligamentous tensions (uncontrolled) in each condition; each measurement for the total knee consisted of 5000 signal averages, and each measurement for the partial knee consisted of 25,000 to compensate for its smaller size (less inductive coupling).

### 2.4. Measurement Sensitivity

The signal-to-noise ratio was lower for the smaller partial knee implant (due to worse inductive coupling with the smaller coil size), and hence, the sensitivity analyses focused on the partial knee implant. A coordinate system that mapped the position and orientation of the transmit coil was defined relative to the intercondylar eminence at the center of the knee (Figure 2). Translations in x, y, and z were in the posteroanterior (PA), mediolateral (ML), and caudocranial (CC) directions, respectively, whilst rotations in ϕ and ψ were angular deviations of the reader’s coil axis about the x and y axes (i.e., in the sagittal and coronal planes), respectively. Rotation about the z axis was not investigated as the reader coil was axisymmetric. The position of the reader coil was varied from the neutral position (x,y,z,ϕ,ψ=0) in each direction independently (range: ±10 mm/±10°; interval: 5 mm/5°) with fifty measurements recorded in each pose prior to fixation of the tibial component. These ranges were set according to the physical possibility of deviations for the Ø105 mm reader. The femoral component was maintained at 30° flexion and no tissue was present.

Sensitivity to musculoskeletal tissue was established by comparing fifty measurements in the neutral position with and without a 10-millimetre-thick wall of porcine soft tissue lining the inside of the reader. The effect of the femoral component on tibial measurements was then analysed by removing the femoral component and acquiring fifty measurements in the neutral reader position. Each of the fifty measurements in the sensitivity studies consisted of 5000 signal averages.

### 2.5. Statistical Analysis

One-tail independent samples t-tests were used to determine statistical significance in the debonding study. Sensitivity to the reader positioning in each measurement direction was analysed independently to determine the potential for a false-positive detection of debonding. The effect on noise (variance) due to displacement was assessed using Levene’s test for the equality of variance in each case, whilst drift (mean difference) was assessed using one-way Welch ANOVAs (a robust test for the equality of means; no assumption of homogeneity of variance). In the anatomical and femoral component sensitivity studies, Levene’s test and two-tail independent samples t-tests were used to determine equality of variance and mean value, respectively. Minimum sample sizes were determined a priori using a power analysis (see Appendix A).

## 3. Results

### 3.1. Debonding Detection with Smaller Partial Knee Replacement

Compared to the well-fixed partial knee implant, the mean relative amplitude between successive echoes increased in all loosening states (Figure 3): partially debonded beneath the transducer (+7.1%; 95% CI: +6.6 to +7.6%; *p* < 0.001), fully debonded in situ (+6.3%; 95% CI: +6.0% to +6.6%; *p* < 0.001), and fully debonded with displacement (+6.4%; 95% CI: +6.0% to +6.9%; *p* < 0.001). A similar trend was seen in the mean relative amplitude measurements before and after cementation (+5.1%; 95% CI: 4.6% to 5.6%; *p* < 0.001). Furthermore, measurement sets were distinct—no crossover in values—for each of the loosening states with respect to the well-fixed implant.

### 3.2. Debonding Detection with Total Knee Replacement

Debonding detection results were similar for the total knee replacement but with an improved signal-to-noise ratio compared to the smaller partial knee replacement. The mean relative amplitude between successive echoes (Γ) was 0.5938 ± 0.0021 when the implant was cemented, rising to 0.7876 ± 0.0019 (+32.6%; *p* < 0.001) and 0.7865 ± 0.0017 (32.5%; *p* < 0.001) when partially and fully debonded, respectively. Again, there was no crossover in values between the well-fixed and debonded states.

### 3.3. Measurement Sensitivity for the Smaller Partial Knee Replacement

Fixation measurements for the smaller partial knee implant were robust to changes in reader positioning in all translational and rotational directions within the ranges investigated (Figure 4; ±10 mm and ±10°). There was homogeneity of variances in the fixation measurements when the reader was displaced in the posteroanterior, sagittal, and coronal directions but not in the mediolateral and caudocranial directions (Table 1). The SNR in the neutral position was 35.4 dB with 5000 signal averages; the best SNR (lowest variance; 38.2 dB) was observed at +10 mm in mediolateral translation where the implant was in closest proximity to the reader; and the worst SNR (highest variance: 31.2 dB) was recorded at +10 mm caudocranial translation when the implant and the reader had the greatest planar displacement. No significant effect was detected on mean fixation measurement due to changes in reader positioning in any direction (Table 1). SNR improved with the number of signal averages per measurement (Figure 5).

Measurements of fixation were also shown to be highly robust to the presence/absence of soft tissue (Figure 6; ΔSNR: −0.72 dB) and the metallic femoral component (Figure 6; ΔSNR: −1.29 dB). The mean differences in fixation measurement due to the presence/absence of soft tissue (−0.001; 95% CI: −0.006 to +0.004) and the metallic femoral component (−0.002; 95% CI: −0.007 to +0.003) were not statistically significant at the level powered (Table 1).

## 4. Discussion

This study evaluated the sensing performance of a low-cost concept for detecting implant–cement debonding following knee arthroplasty. Even at the small scale of the partial knee implant, it was found that tibial debonding could be robustly detected when the implant was partially debonded, fully debonded in situ, and fully debonded with displacement (all *p* < 0.001) under varying degrees of ligamentous tension and femoral component position. This measurement was not affected by the positioning of the external reader or femoral component, and musculoskeletal tissue was quantified. To our knowledge, this is the first study to successfully embody a loosening detection method for a small partial knee replacement implant.

Clinical implementation of the technology is attractive as the functionalisation of the implants does not require modification of their external form. Thus, the smart implants were able to be implanted with the standard instrument set and surgical technique. This would minimise cost upon translation as it would not increase hospital inventory for surgical instruments and would not require surgeons to learn new operative techniques. The form factor of the modification was also sufficiently small to apply the concept to other implants, including hip and shoulder components, with loosening remaining a key challenge for the latter [38,39]. The low cost of the embedded piezoelectric transducer (<USD 1) removes barriers to translation into regular clinical practice, which have beset previous smart implants.

The coil arrangement in the present study allowed improved inductive coupling compared to our previous work [36]; with the new arrangement, displacements of 10 mm had little effect on the loosening measurement (Figure 4). This is a significant improvement over our previous work, where we were not able to measure loosening at this distance. Other authors have proposed modified implants to detect loosening [33,35,40,41,42,43], most of which were also evaluated under simulated conditions. Of those concepts, an acoustic analysis technique was able to detect osseointegration in rabbits in vivo [44]. Their excitation and acquisition were also percutaneous at a depth < 2 mm, which was considerably less than the implantation depth simulated here (tissue depth > 10 mm).

Synthetic rather than cadaveric bone was used. This limitation was considered acceptable as the fixation measurement is primarily driven by the implant–cement interface, not the cement–bone interface. The cement–implant interface is designed to withstand millions of load cycles, and hence, inducing loosening through cyclic loading was impractical. Rather, a lift-off moment was used to simulate a mechanism for aseptic loosening described in the literature [45,46,47]. It was sufficiently controllable to enable research for a partial loosening state, and it produced a failure at the implant–cement interface that resembled contemporary clinical reports of tibial debonding [7,8,9,10,11,12,13,14,15,16,17,18]. The work is also limited in that only a single cementation technique for a single brand of bone–cement was studied. The cementation technique does vary between surgeons and manufacturers, and exact failure mechanisms and timescales are yet to be fully described; therefore, it is challenging to prescribe a sensitivity and specificity for tibial debonding detection without clinical validation. It is expected that a single embedded transducer would suffice for a tibial-debonding mechanism characterised by fast-acting catastrophic failure, whereas a slow-acting propagation mechanism would necessitate an array of embedded piezoelectric transducers.

Clinical implementation of this technology would need to consider data security (the implanted device stores no data, but the external reader data would need to be uploaded to a hospital computer system). Also, while piezoelectric transducers are widely regarded as appropriate for long-term structural health monitoring applications [48], their use for long-term monitoring in an implant would require further research. For example, to verify that the transducer would not be damaged in the event of extreme implant-bearing wear. An advantage of the developed solution is that by minimising the number of electronics implanted, such testing is greatly simplified. Finally, the piezoelectric transducer used was lead-based (PZT). While in the short-term, this material does not lead to cytotoxic effects for bone cells [49], long-term lead ion release is a concern, and hence, the sensor would likely require hermetical sealing for clinical applications. In our previous research, we also demonstrated that lead-free alternatives, such as BNT-6BT, could be used to acquire ultrasonic pulse-echo measurements in applications where hermetic sealing is not possible [49].

## 5. Conclusions

In this study, it was demonstrated that tibial debonding at the implant–cement interface of the smallest clinically relevant knee implant, a fixed lateral partial knee, could be reliably detected in a laboratory model of orthopaedic surgery and postoperative loosening. Debonding results were then replicated on a larger total knee implant in the same model. The novel system was insensitive to the reader positioning and tissue and adjacent large metallic components. With its low cost and small form factor, the smart implant concept could be developed towards clinical trials to enable new research into uncertain aseptic loosening mechanisms, with the potential for future use in clinical practice to inform decision making.

## Figures and Tables

**Figure 1 sensors-24-01696-f001:**
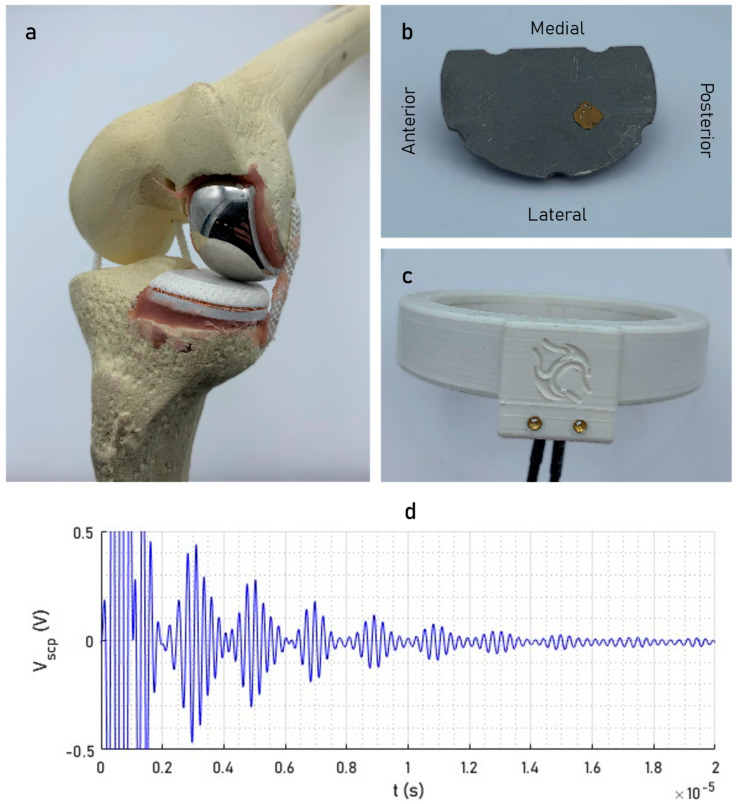
The instrumented unicondylar knee prosthesis (**a**) shown cemented into a synthetic bone model and (**b**) showing the position of the shear-wave piezoelectric transducer on the tibial component; (**c**) the reader used to acquire wireless pulse-echo measurements; and (**d**) the pulse-echo response.

**Figure 2 sensors-24-01696-f002:**
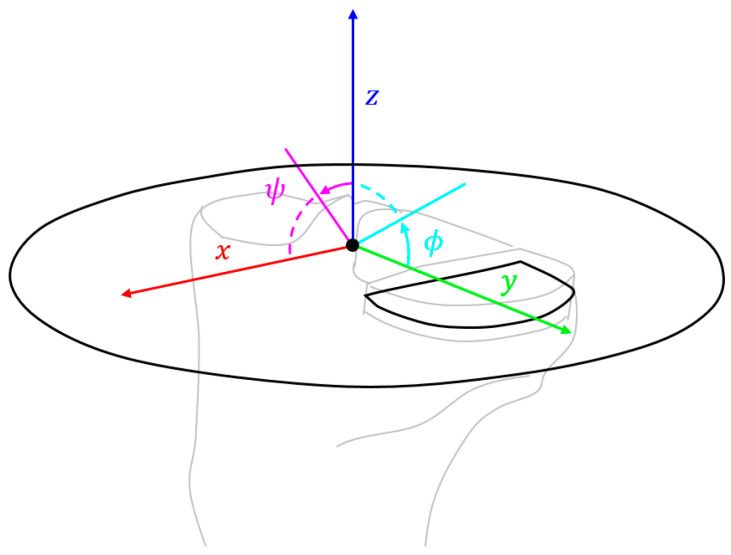
The coordinate system centred on the intercondylar eminence of the tibia used to describe the position and orientation of the reader–embedded coil and external transmitter coil in black.

**Figure 3 sensors-24-01696-f003:**
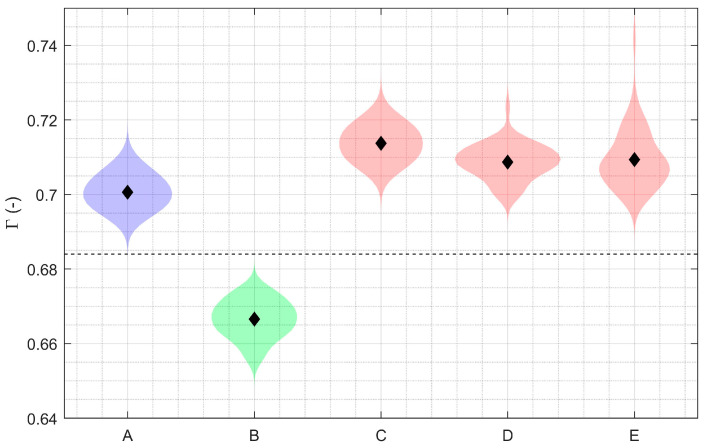
Violin plots showing a decrease in the fixation parameter (Γ) when the uncemented partial knee implant (A; blue) was cemented into synthetic bone (B; green), and then a subsequent increase when the implant–cement interface was compromised beneath the piezoelectric transducer (red): partially debonded (C), fully debonded in situ (D), and fully debonded with gross displacement (E). The diamond symbols indicate the mean, and the shaded regions are the distribution of the data.

**Figure 4 sensors-24-01696-f004:**
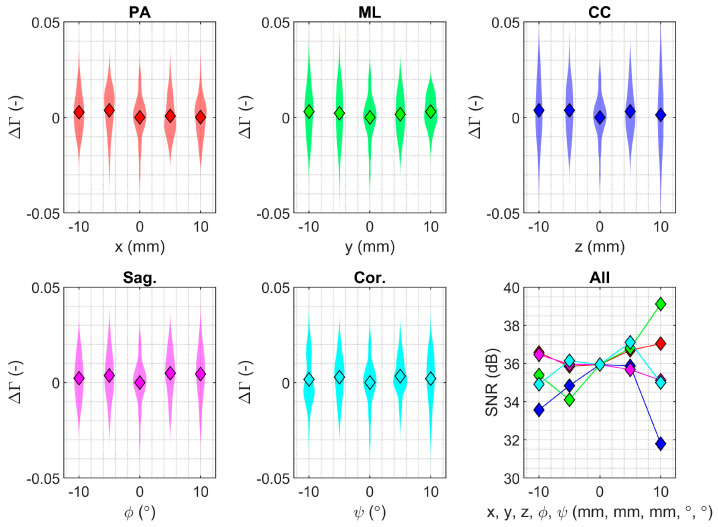
Deviation in the fixation measurement (Γ) from the mean value recorded in the neutral position, shown as violin plots, for anteroposterior translation (red, **top left**), mediolateral translation (green, **top middle**), craniocaudal translation (dark blue, **top right**), sagittal plane rotation (pink, **bottom left**) and coronal plane rotation (light blue, **bottom middle**), and corresponding signal-to-noise ratio plot (**bottom right**, colours consistent with that of the individual plots). The diamond symbols indicate the mean, and the shaded regions are the distribution of the data.

**Figure 5 sensors-24-01696-f005:**
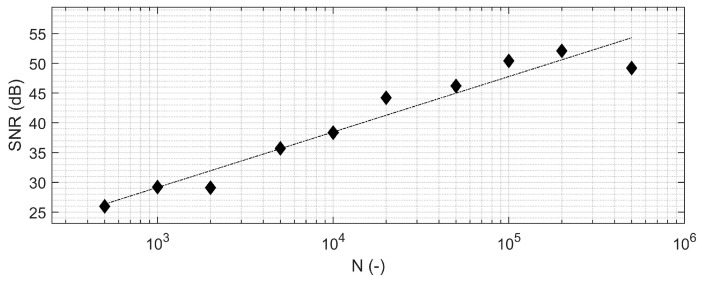
Signal-to-noise (SNR) for the fixation measurement (Γ) increased by ~20 dB/decade as a function of the number of signal averages.

**Figure 6 sensors-24-01696-f006:**
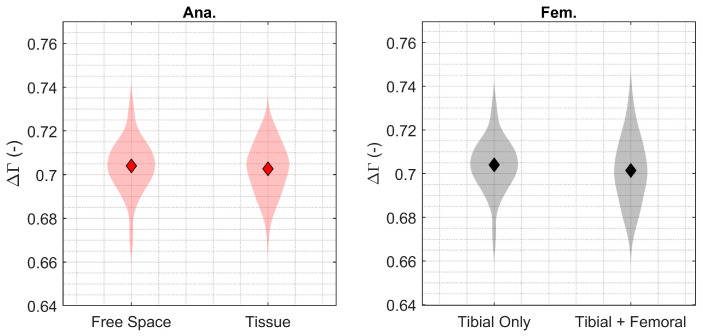
Sensitivity of the fixation parameter measurement (Γ) to transmission through free space versus soft tissue (Ana.: **left**, red) and the presence of the femoral component (Fem.: **right**, grey), shown as violin plots. The diamond symbols indicate the mean, and the shaded regions are the distribution of the data.

**Table 1 sensors-24-01696-t001:** Statistical analyses for the difference in fixation measurement due to reader positioning in posteroanterior (PA), mediolateral (ML), caudocranial (CC), sagittal (Sag.), and coronal (Cor.), and due to the presence of the femoral component (Fem.), and soft tissue (Ana.).

	Homogeneity of Variance		Equality of Mean	
	Stat.	Sig.	Stat.	Sig.
PA	F(4,245) = 0.354	*p* = 0.841	F(4,122.413) = 1.187	*p* = 0.329
ML	F(4,245) = 3.264	*p* = 0.012	F(4,121.398) = 0.596	*p* = 0.666
CC	F(4,245) = 4.563	*p* = 0.001	F(4,121.816) = 0.898	*p* = 0.467
Sag.	F(4,245) = 0.501	*p* = 0.735	F(4,122.397) = 1.361	*p* = 0.252
Cor.	F(4,245) = 1.469	*p* = 0.212	F(4,122.189) = 0.591	*p* = 0.670
Fem.	F(1,98) = 2.219	*p* = 0.140	t(95.974) = 0.599	*p* = 0.551
Ana.	F(1,98) = 0.000	*p* = 0.999	t(97.310) = 0.435	*p* = 0.664

## Data Availability

Analyzed data are contained within the article. Raw data are available upon reasonable request.

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
