# Peer review of "Passive Biotelemetric Detection of Tibial Debonding in Wireless Battery-Free Smart Knee Implants"

_sensors, 2024, doi:10.3390/s24051696_

Round 1

Reviewer 1 Report

Comments and Suggestions for Authors

I suggest Minor revision prior to the acceptance

In abstract portion please include key results of debonding detection and the relation to sensor application.

In introduction section please establish state-of-the-art by citing paper on tibial deboding and wireless detection. How the work stated in this work is differencet compared to the literature. The novelty statement should be included.

In simulated tibial debonding and  mesurement sensitiveity please correspond the sensor efficency with the relevant papers and the difference in results should be justified.

The quality of the Figure 1 is very poor and should be improved.

Furthermore, the methodolgy sectiuon should iterate that how the partiulare sensor was rated and software analysis should be in detail. Please include each and every step involved in programming and simulation process

Author Response

I suggest Minor revision prior to the acceptance

Comment #1: In abstract portion please include key results of debonding detection and the relation to sensor application.

Author response: We have edited the abstract as advised:

“The sensing system comprised of a piezoelectric transducer and coil, with the transducer affixed to the superior surface of the tibial trays of both partial (PKR) and total knee replacement (TKR) systems. Fixation was measured via pulse-echo responses elicited via a three-coil inductive link. The instrumented systems could detect loss of fixation when the implants were partially debonded (+7.1% PKA, +32.6% TKA, both p < 0.001) and fully debonded in situ (+6.3% PKA, +32.5% TKA, both p < 0.001).”

Comment #2: In introduction section please establish state-of-the-art by citing paper on tibial deboding and wireless detection. How the work stated in this work is differencet compared to the literature. The novelty statement should be included.

Author response. We have added an additional paragraph to the introduction which reads:

In our previous work [36], a concept for loosening detection requiring only two implanted components (a piezoelectric transducer and a coil) was demonstrated in benchtop tests. The aim of this work was to embody the system into modern knee replacement systems and comprehensively characterise measurement sensitivity for detecting tibial debonding. A partial knee prothesis was analysed as it is one of the smallest cemented implants in routine clinical use, and to-date, no smart implant system has been small enough to be successfully embodied within such a device. A total knee replacement system was also analysed as it is the highest volume cemented orthopaedic procedure.”

We then re-iterate the novelty of our new work in the first paragraph of the discussion which now reads:

“To our knowledge, this is the first study to successfully embody a loosening detection method for a small partial knee replacement implant.”

Comment #3: In simulated tibial debonding and  mesurement sensitiveity please correspond the sensor efficency with the relevant papers and the difference in results should be justified.

Author response: As a stylistic choice, we prefer to separate our results and discussion, so we have addressed your comment in our discussion section, rather than the result section. Our discussion now reads:

The coil arrangement in the present study allowed improved inductive coupling compared to our previous work [36]; with the new arrangement, displacements of 10 mm had little effect on the loosening measurement (Fig. 4.). This is a significant improvement over our previous work where we were not able to measure loosening at this distance. Other authors have proposed modified implants to detect loosening [33, 35, 40-43], most of which were also evaluated under simulated conditions. Of those concepts, an acoustic analysis technique was able to detect osseointegration in rabbits in vivo [44]. Their excitation and acquisition were also percutaneous at a depth < 2 mm, which was considerably less than the implantation depth simulated here (tissue depth > 10 mm).”

Comment #4: The quality of the Figure 1 is very poor and should be improved.

Author response: We have uploaded a high-resolution version of this image.

Comment #5: Furthermore, the methodolgy sectiuon should iterate that how the partiulare sensor was rated and software analysis should be in detail. Please include each and every step involved in programming and simulation process

Authors response: Thank you for this suggestion. We realise now that our software and data analysis was poorly signposted. We have edited it to give the software analysis its own paragraph. We believe that it is now described in sufficient detail that it can be replicated. However, we would also be happy to provide the code for the data analysis if it is still not clear; please let us know if you would like us to do so as supplementary material. The manuscript methods now read:

Raw signals were initiated by a five-cycle, Hanning-windowed toneburst (central frequency: 3.5 MHz) and the response was sampled at 100 MHz (Handyscope HS5, TiePie, Sneek, Netherlands). These signals were amplified (gain: 60 dB; WaveMaker Duet, Macro Design Ltd., London, UK).

Data were then analysed in MatLab (R2018b, MathWorks) with a custom script. The data were: averaged across many measurements (N > 5000), bandpass-filtered (passband: 2-5 MHz), resampled at 10 GHz, and cross-correlated with the original signals. Hilbert envelopes were extracted from the processed signals and the fixation measurement () was computed as the arithmetic average of the relative amplitude between successive echoes for the first three echo pairs.”

Reviewer 2 Report

Comments and Suggestions for Authors

This study provides valuable insights into the predominant failure mechanism in contemporary knee replacement surgery, namely aseptic loosening, and proposes an innovative solution in the form of smart implants. The research clearly delineates the existing challenges in the identification of early-stage loosening and differentiation between aseptic loosening and infections using current diagnostic techniques, establishing a compelling rationale for the study. The introduction of a novel approach involving smart implants for monitoring demonstrates innovation, characterized by a wireless, battery-free design that effectively reduces device costs, size, and associated risks.The cost-effectiveness and compact size of the smart implants render them highly suitable for widespread clinical applications, particularly in investigating the mechanisms of aseptic loosening where uncertainties exist. However, there are still some minor issues that need to be addressed.

1. To enhance the credibility of the proposed wireless, battery-free smart implants in real patients, it is crucial to incorporate validation with actual clinical cases. This validation aims to ensure the feasibility and effectiveness of the smart implants in authentic clinical scenarios, thereby bolstering the reliability of the research findings.

2. Further consideration should be given to the long-term monitoring capabilities of the implants. This is essential for detecting changes in the implant's status, particularly during extended periods of use. Understanding the long-term stability of the implants and their early sensitivity to loosening becomes imperative for comprehensive assessment.

3. A comprehensive evaluation of the performance and advantages of the proposed techniques in detecting loosening should be conducted through a comparative analysis with existing methods used in current clinical practices. This comparative approach will help ascertain the superiority of the proposed method over existing technologies.

4. Conduct a thorough analysis of the biocompatibility between the implant and surrounding tissues to ensure that it does not elicit allergic reactions or other adverse effects. Addressing biocompatibility is critical for the successful clinical application of the technology.

5. Given the use of sensors and communication components, careful consideration must be given to data security and privacy issues. Clearly articulate the measures taken in the study to safeguard the privacy and security of patient data.

6. Explore the feasibility of integrating the smart implants with other medical devices to enhance overall health monitoring for patients. This integration has the potential to contribute to more comprehensive treatment and management strategies.

7. Investigate the applicability of the implants across different types of knee replacement surgeries to ensure accurate monitoring and diagnosis in diverse contexts.

8. Conduct a more detailed statistical analysis of the measurement results mentioned in the study, including sensitivity, specificity, and predictive values, to comprehensively assess the performance of the implants.

9. Collaborate closely with clinical doctors and surgical teams to ensure that the proposed technology can be effectively used in real medical settings and seamlessly integrated into routine surgical practices.

10. In the conclusion of the research, discuss potential avenues for future improvements, including new sensing technologies, data processing methods, or other innovations that could enhance the system's performance. Most recent studies such as DOI: 10.12336/biomatertransl.2022.03.002; DOI: 10.3877/cma.j.issn.2096-112X.2020.01.009; doi.org/10.1016/j.puhe.2023.10.022 are recommended to be cited in proper places.

Author Response

Comment #1. “To enhance the credibility of the proposed wireless, battery-free smart implants in real patients, it is crucial to incorporate validation with actual clinical cases. This validation aims to ensure the feasibility and effectiveness of the smart implants in authentic clinical scenarios, thereby bolstering the reliability of the research findings.”

Author response: We agree – this would be a truly impactful addition. In the field of knee replacement surgery, no smart implants are indicated for clinical decision making. Such an implant would be a class III medical device in Europe (where we are based) and in the US (where the manufacturer we are collaborating with is headquartered). To take such a device to clinical trial would require more than $5M invested, and hence fall outside of the scope of what can be achieved in our lab-based proof-of-concept study.

Even without clinical trial, a significant advance is presented in the study – we have proven for the first time that it is possible to add sensors to partial knee replacement implants and acquire meaningful data about the fixation of the implant, with readings insensitive to the common hurdles (such as reader location, and noise created by large metallic components adjacent to the inductive coils). We believe these advance warrants publication to move the field forwards and build track record towards our ultimate goal of a clinical trial for patient benefit.

We have edited our conclusion to highlight this in more detail. It now reads:

In this study, it was demonstrated that tibial debonding at the implant-cement interface of the smallest clinically relevant knee implant, a fixed lateral partial knee, could be reliably detected in laboratory model of orthopaedic surgery and postoperative loosening. Debonding results were then replicated on a larger total knee implant in the same model. The novel system was insensitive to reader positioning, tissue and adjacent large metallic components. With its low cost and small form factor, the smart implant concept could be developed towards clinical trial to enable new research into uncertain aseptic loosening mechanisms, with the potential for future use in clinical practice to inform decision making.”

  1. Further consideration should be given to the long-term monitoring capabilities of the implants. This is essential for detecting changes in the implant's status, particularly during extended periods of use. Understanding the long-term stability of the implants and their early sensitivity to loosening becomes imperative for comprehensive assessment.

We now discuss long-term use of the sensors in our article:

“While piezoelectric transducers are widely regarded as appropriate for long-term structural health monitoring applications [48], their use for long-term monitoring in an implant would require further research. For example, to verify that the transducer would not be damaged in the event of extreme implant bearing wear. An advantage of the developed solution is that by minimising the amount of electronics implanted, such testing is greatly simplified.”

  1. A comprehensive evaluation of the performance and advantages of the proposed techniques in detecting loosening should be conducted through a comparative analysis with existing methods used in current clinical practices. This comparative approach will help ascertain the superiority of the proposed method over existing technologies.

The proposed technology is highly novel, with no equivalent technology for detecting loosening either on the market or under clinical trial. Canary Medical launched the first commercially available smart knee replacement device in 2023. This implant measures kinematics and is not indicated for loosening.

Thus, rather than compare to clinical technology, we instead compare to other advances in the research domain. Our discussion reads:

“Other authors have proposed modified implants to detect loosening [33, 35, 40-43], most of which were also evaluated under simulated conditions. Of those concepts, an acoustic analysis technique was able to detect osseointegration in rabbits in vivo [44]. Their excitation and acquisition were also percutaneous at a depth < 2 mm, which was considerably less than the implantation depth simulated here (tissue depth > 10 mm).”

  1. Conduct a thorough analysis of the biocompatibility between the implant and surrounding tissues to ensure that it does not elicit allergic reactions or other adverse effects. Addressing biocompatibility is critical for the successful clinical application of the technology.

We comprehensively analysed its biocompatibility in our previous research, and we now highlight this in our discussion.

“Finally, the piezoelectric transducer used was lead-based (PZT). While in the short-term this material does not lead to cytotoxic effects for bone cells [48], long-term lead ion release is a concern and hence the sensor would likely require hermetical sealing for clinical applications. In our previous research, we also demonstrated that lead-free alternatives, such as BNT-6BT, could be used to acquire ultrasonic pulse-echo measurements in applications where hermetic sealing is not possible [48].”

  1. Given the use of sensors and communication components, careful consideration must be given to data security and privacy issues. Clearly articulate the measures taken in the study to safeguard the privacy and security of patient data.

No data is stored on the implant which enhances data security. We have added a note to the discussion to highlight that data security for the reader would need to be considered in future work. It now reads:

“A clinical implementation of this technology would need to consider data security (the implanted device stores no data, but the external reader data would need to be uploaded to a hospital computer system).”

  1. Explore the feasibility of integrating the smart implants with other medical devices to enhance overall health monitoring for patients. This integration has the potential to contribute to more comprehensive treatment and management strategies.

Thank you for this suggestion. It is an interesting expansion for our work. We will consider this going forwards; however it falls outside the scope of the present study which considers loosening following knee replacement.

  1. Investigate the applicability of the implants across different types of knee replacement surgeries to ensure accurate monitoring and diagnosis in diverse contexts.

This is a key advantage of our work, we not only considered total knee arthroplasty (which others have done), but also considered partial knee arthroplasty. We now highlight this in our introduction:

“A partial knee prothesis was analyzed as it is one of the smallest cemented implants in routine clinical use, and to-date, no smart implant system has been small enough to be successfully embodied within such a device.”

  1. Conduct a more detailed statistical analysis of the measurement results mentioned in the study, including sensitivity, specificity, and predictive values, to comprehensively assess the performance of the implants.

All differences are reported with corresponding statistical analysis. E.g. the results read:

“Compared to the well-fixed partial knee implant, the mean relative amplitude between successive echoes increased in all loosening states (Figure 3): partially debonded beneath the transducer (+7.1%; 95% CI: +6.6 to +7.6%; p < 0.001), fully debonded in situ (+6.3%; 95% CI: +6.0% to +6.6%; p < 0.001), and fully debonded with displacement (+6.4%; 95% CI: +6.0% to +6.9%; p < 0.001).”

We have also included a comprehensive power analysis as an Appendix which reads:

“For the debonding study, mean (μ = 0.706) and standard deviation (σ = 0.0059) for the debonded implant were calculated from fifty measurements of the implant before cementation. The mean difference (−15%) due to cementation was estimated based on debonding detection results from a previous study [36] whilst variance was calculate from 50 pre-cementation measurements and was assumed homogeneous for the purpose of calculation. Target significance (α) and power (1-β) were set to standard values of 5% and 90% respectively, and the allocation ratio (well-fixed vs. debonded) was set as even. The minimum sample size (n > 2 per group) was computed using G*Power 3.1 software [49]; higher sample sizes (n = 50 per group) were used in the eventual experimental protocol (actual power > 99.9%).

For the reader positioning sensitivity study, the same mean was used as the basis for the loosening power calculation, but a higher standard deviation was used to account for the lower number of signal averages (σ = 0.01319). The clinically important mean difference that would trigger a false positive for loosening was drawn from the debonding detection results (Figure 3; ε = 0.049). Target significance (α) and power (1-β) were again set to 5% and 90% respectively, and allocation ratios were set as even. For calculation purposes, the mean vector was set up to detect a clinically important mean difference at the extent of the displacement range, where standard deviation was assumed to increase by 10%, i.e.

μ=(μ, μ, μ, μ, μ+ε)            (1)

and

σ= (1.1σ, 1.05σ, σ, 1.05σ, 1.1σ).              (2)

Minimum sample sizes were computed using an R program developed by Jan and Shieh [50]. The minimum sample size for loosening (n > 4 per group) was lower than the 50 measurements were ultimately acquired per group.

The same mean (μ), standard deviation (σ), clinically important mean difference (ε), significance (α), and power (1-β) were used for the other sensitivity studies. Minimum samples were calculated using G*Power 3.1 software for debonding (n > 3 per group). 50 were ultimately acquired per group.”

With regards to sensitivity, specificity and area under the curve. In a bench top model, it is not possible to measure these clinical parameters, and hence it falls outside of the scope of our article, as per our response to comment #1.

  1. Collaborate closely with clinical doctors and surgical teams to ensure that the proposed technology can be effectively used in real medical settings and seamlessly integrated into routine surgical practices.

Thank you for highlighting this, we realised we did not acknowledge the clinicians who provided insight for our work. We have now amended this:

“Thank you also to Mr Gareth Jones FRCS and Mr Omar Musbahi for sharing their clinical insight, and for providing access to a patient and public involvement group, from who we also received invaluable feedback.”

  1. In the conclusion of the research, discuss potential avenues for future improvements, including new sensing technologies, data processing methods, or other innovations that could enhance the system's performance. Most recent studies such as DOI: 10.12336/biomatertransl.2022.03.002; DOI: 10.3877/cma.j.issn.2096-112X.2020.01.009; doi.org/10.1016/j.puhe.2023.10.022 are recommended to be cited in proper places.

We added a key reference with regards to long-term stability of piezoelectric transducer for structural health monitoring, bringing our total reference list to 51. Of these, 49 are from research groups other than our own, and hence we believe we have adequately referenced the works of others in the field.

Thank you for the suggested articles. We reviewed them in detail and found them interesting reads. We have chosen to not cite them in this Sensors article, as they are not directly relevant to the work under consideration:

10.12336/biomatertransl.2022.03.002 explores mesenchymal stem cell-derived extracellular vesicles as a possible therapeutic strategy for orthopaedic diseases. Cell biology basic science is a very different topic and falls outside of the scope of an article targeted at the Sensors journal audience.

10.3877/cma.j.issn.2096-112X.2020.01.009 explores hyaluronic acid-based hydrogels with tobacco mosaic virus for bone repair. Biologics are a very different topic, and falls outside of the scope of an article targeted at the Sensors journal audience.

doi.org/10.1016/j.puhe.2023.10.022 concludes that osteoporosis is associated with depression among older adults. The main indication for knee replacement is osteoarthritis; osteoporosis is not currently an indication for knee replacement.